# Adaptive Sampling
# Towards Fast Graph Representation Learning

**Wenbing Huang**[1]**, Tong Zhang**[2]**, Yu Rong**[1]**, Junzhou Huang**[1]
[1] Tencent AI Lab. ;
[2] Australian National University;
hwenbing@126.com, tong.zhang@anu.edu.au,
yu.rong@hotmail.com, joehhuang@tencent.com

## Abstract

Graph Convolutional Networks (GCNs) have become a crucial tool on learning representations of graph vertices. The main challenge of adapting GCNs on large-scale graphs is the scalability issue that it incurs heavy cost both in computation and memory due to the uncontrollable neighborhood expansion across layers. In this paper, we accelerate the training of GCNs through developing an adaptive layer-wise sampling method. By constructing the network layer by layer in a top-down passway, we sample the lower layer conditioned on the top one, where the sampled neighborhoods are shared by different parent nodes and the over expansion is avoided owing to the fixed-size sampling. More importantly, the proposed sampler is adaptive and applicable for explicit variance reduction, which in turn enhances the training of our method. Furthermore, we propose a novel and economical approach to promote the message passing over distant nodes by applying skip connections. Intensive experiments on several benchmarks verify the effectiveness of our method regarding the classification accuracy while enjoying faster convergence speed.

## 1  Introduction

Deep Learning, especially Convolutional Neural Networks (CNNs), has revolutionized various machine learning tasks with grid-like input data, such as image classification [1] and machine translation [2]. By making use of local connection and weight sharing, CNNs are able to pursue translational invariance of the data. In many other contexts, however, the input data are lying on irregular or non-euclidean domains, such as graphs which encode the pairwise relationships. This includes examples of social networks [3], protein interfaces [4], and 3D meshes [5]. How to define convolutional operations on graphs is still an ongoing research topic.

There have been several attempts in the literature to develop neural networks to handle arbitrarily structured graphs. Whereas learning the graph embedding is already an important topic [6, 7, 8], this paper mainly focus on learning the representations for graph vertices by aggregating their features/attributes. The closest work to this vein is the Graph Convolution Network (GCN) [9] that applies connections between vertices as convolution filters to perform neighborhood aggregation. As demonstrated in [9], GCNs have achieved the state-of-the-art performance on node classification.

An obvious challenge for applying current graph networks is the scalability. Calculating convolutions requires the recursive expansion of neighborhoods across layers, which however is computationally prohibitive and demands hefty memory footprints. Even for a single node, it will quickly cover a large portion of the graph due to the neighborhood expansion layer by layer if particularly the graph is dense or powerlaw. Conventional mini-batch training is unable to speed up the convolution computations, since every batch will involve a large amount of vertices, even the batch size is small.

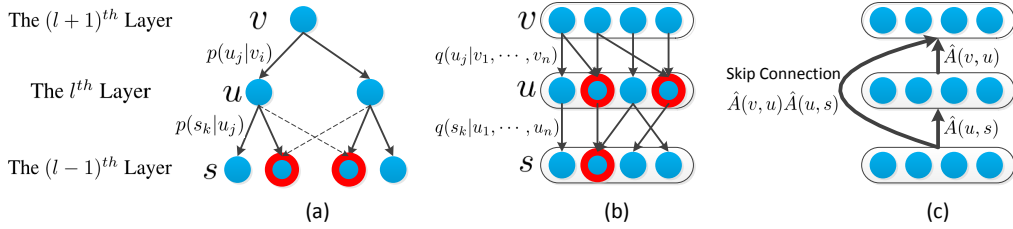

The $(l+1)^{th}$ Layer

The $l^{th}$ Layer

The $(l-1)^{th}$ Layer

$p(u_j|v_i)$

$p(s_k|u_j)$

$q(u_j|v_1, \cdots, v_n)$

$q(s_k|u_1, \cdots, u_n)$

Skip Connection

$\hat{A}(v,u)\hat{A}(u,s)$

$\hat{A}(v,u)$

$\hat{A}(u,s)$

(a)        (b)        (c)

Figure 1: Network construction by different methods: (a) the node-wise sampling approach; (b) the layer-wise sampling method; (c) the model considering the skip-connection. To illustrate the effectiveness of the layer-wise sampling, we assume that the nodes denoted by the red circle in (a) and (b) have at least two parents in the upper layer. In the node-wise sampling, the neighborhoods of each parent are not seen by other parents, hence the connections between the neighborhoods and other parents are unused. In contrast, for the layer-wise strategy, all neighborhoods are shared by nodes in the parent layer, thus all between-layer connections are utilized.

To avoid the over-expansion issue, we accelerate the training of GCNs by controlling the size of the sampled neighborhoods in each layer (see Figure 2). Our method is to build up the network layer by layer in a top-down way, where the nodes in the lower layer[1] are sampled conditionally based on the upper layer's. Such layer-wise sampling is efficient in two technical aspects. First, we can reuse the information of the sampled neighborhoods since the nodes in the lower layer are visible and shared by their different parents in the upper layer. Second, it is easy to fix the size of each layer to avoid over-expansion of the neighborhoods, as the nodes of the lower layer are sampled as a whole.

The core of our method is to define an appropriate sampler for the layer-wise sampling. A common objective to design the sampler is to minimize the resulting variance. Unfortunately, the optimal sampler to minimize the variance is uncomputable due to the inconsistency between the top-down sampling and the bottom-up propagation in our network (see § 4.2 for details). To tackle this issue, we approximate the optimal sampler by replacing the uncomputable part with a self-dependent function, and then adding the variance to the loss function. As a result, the variance is explicitly reduced by training the network parameters and the sampler.

Moreover, we explore how to enable efficient message passing across distant nodes. Current methods [6, 10] resort to random walks to generate neighborhoods of various steps, and then take integration of the multi-hop neighborhoods. Instead, this paper proposes a novel mechanism by further adding a skip connection between the $(l+1)$-th and $(l-1)$-th layers. This short-cut connection reuses the nodes in the $(l-1)$-th layer as the 2-hop neighborhoods of the $(l+1)$-th layer, thus it naturally maintains the second-order proximity without incurring extra computations.

To sum up, we make the following contributions in this paper: **I.** We develop a novel layer-wise sampling method to speed up the GCN model, where the between-layer information is shared and the size of the sampling nodes is controllable. **II.** The sampler for the layer-wise sampling is adaptive and determined by explicit variance reduction in the training phase. **III.** We propose a simple yet efficient approach to preserve the second-order proximity by formulating a skip connection across two layers. We evaluate the performance of our method on four popular benchmarks for node classification, including Cora, Citeseer, Pubmed [11] and Reddit [3]. Intensive experiments verify the effectiveness of our method regarding the classification accuracy and convergence speed.

## 2   Related Work

While graph structures are central tools for various learning tasks (*e.g.* semi-supervised learning in [12, 9]), how to design efficient graph convolution networks has become a popular research topic. Graph convolutional approaches are often categorized into spectral and non-spectral classes [13]. The spectral approach first proposed by [14] defines the convolution operation in Fourier domain. Later, [15] enables localized filtering by applying efficient spectral filters, and [16] employs Chebyshev

expansion of the graph Laplacian to avoid the eigendecomposition. Recently, GCN is proposed in [9] to simplify previous methods with first-order expansion and re-parameterization trick. Non-spectral approaches define convolution on graph by using the spatial connections directly. For instance, [17] learns a weight matrix for each node degree, the work by [18] defines multiple-hop neighborhoods by using the powers series of a transition matrix, and other authors [19] extracted normalized neighborhoods that contain a fixed number of nodes.

A recent line of research is to generalize convolutions by making use of the patch operation [20] and self-attention [13]. As opposed to GCNs, these methods implicitly assign different importance weights to nodes of a same neighborhood, thus enabling a leap in model capacity. Particularly, Monti *et al.* [20] presents mixture model CNNs to build CNN architectures on graphs using the patch operation, while the graph attention networks [13] compute the hidden representations of each node on graph by attending over its neighbors following a self-attention strategy.

More recently, two kinds of sampling-based methods including GraphSAGE [3] and FastGCN [21] were developed for fast representation learning on graphs. To be specific, GraphSAGE computes node representations by sampling neighborhoods of each node and then performing a specific aggregator for information fusion. The FastGCN model interprets graph convolutions as integral transforms of embedding functions and samples the nodes in each layer independently. While our method is closely related to these methods, we develop a different sampling strategy in this paper. Compared to GraphSAGE that is node-wise, our method is based on layer-wise sampling as all neighborhoods are sampled as altogether, and thus can allow neighborhood sharing as illustrated in Figure 2. In contrast to FastGCN that constructs each layer independently, our model is capable of capturing the between-layer connections as the lower layer is sampled conditionally on the top one. We detail the comparisons in § 6. Another related work is the control-variate-based method by [22]. However, the sampling process of this method is node-wise, and the historical activations of nodes are required.

## 3 Notations and Preliminaries

**Notations.** This paper mainly focuses on undirected graphs. Let $\mathcal{G} = (\mathcal{V}, \mathcal{E})$ denote the undirected graph with nodes $v_i \in \mathcal{V}$, edges $(v_i, v_j) \in \mathcal{E}$, and $N$ defines the number of the nodes. The adjacency matrix $A \in \mathbb{R}^{N \times N}$ represents the weight associated to edge $(v_i, v_j)$ by each element $A_{ij}$. We also have a feature matrix $X \in \mathbb{R}^{N \times D}$ with $x_i$ denoting the $D$-dimensional feature for node $v_i$.

**GCN.** The GCN model developed by Kipf and Welling [9] is one of the most successful convolutional networks for graph representation learning. If we define $h^{(l)}(v_i)$ as the hidden feature of the $l$-th layer for node $v_i$, the feed forward propagation becomes

$$h^{(l+1)}(v_i) \quad = \quad \sigma \left( \sum_{j=1}^{N} \hat{a}(v_i, u_j) h^{(l)}(u_j) W^{(l)} \right), \quad i = 1, \cdots, N, \tag{1}$$

where $\hat{A} = (\hat{a}(v_i, u_j)) \in \mathbb{R}^{N \times N}$ is the re-normalization of the adjacency matrix; $\sigma(\cdot)$ is a nonlinear function; $W^{(l)} \in \mathbb{R}^{D^{(l)} \times D^{(l-1)}}$ is the filter matrix in the $l$-th layer; and we denote the nodes in the $l$-th layer as $u_j$ to distinguish them from those in the $(l+1)$-th layer.

## 4 Adaptive Sampling

Eq. (1) indicates that, GCNs require the full expansion of neighborhoods for the feed forward computation of each node. This makes it computationally intensive and memory-consuming for learning on large-scale graphs containing more than hundreds of thousands of nodes. To circumvent this issue, this paper speeds up the feed forward propagation by adaptive sampling. The proposed sampler is adaptable and applicable for variance reduction.

We first re-formulate the GCN update to the expectation form and introduce the node-wise sampling accordingly. Then, we generalize the node-wise sampling to a more efficient framework that is termed as the layer-wise sampling. To minimize the resulting variance, we further propose to learn the layer-wise sampler by performing variance reduction explicitly. Lastly, we introduce the concept of skip-connection, and apply it to enable the second-order proximity for the feed-forward propagation.

### 4.1 From Node-Wise Sampling to Layer-Wise Sampling

**Node-Wise Sampling.** We first observe that Eq (1) can be rewritten to the expectation form, namely,

$$h^{(l+1)}(v_i) = \sigma_{W^{(l)}}(N(v_i)\mathbb{E}_{p(u_j|v_i)}[h^{(l)}(u_j)]), \qquad (2)$$

where we have included the weight matrix $W^{(l)}$ into the function $\sigma(\cdot)$ for concision; $p(u_j|v_i) = \hat{a}(v_i, u_j)/N(v_i)$ defines the probability of sampling $u_j$ given $v_i$, with $N(v_i) = \sum_{j=1}^{N} \hat{a}(v_i, u_j)$.

A natural idea to speed up Eq. (2) is to approximate the expectation by Monte-Carlo sampling. To be specific, we estimate the expectation $\mu_p(v_i) = \mathbb{E}_{p(u_j|v_i)}[h^{(l)}(u_j)]$ with $\hat{\mu}_p(v_i)$ given by

$$\hat{\mu}_p(v_i) = \frac{1}{n}\sum_{j=1}^{n} h^{(l)}(\hat{u}_j), \quad \hat{u}_j \sim p(u_j|v_i). \qquad (3)$$

By setting $n \ll N$, the Monte-Carlo estimation can reduce the complexity of (1) from $O(|E|D^{(l)}D^{(l-1)})$ ($|E|$ denotes the number of edges) to $O(n^2 D^{(l)}D^{(l-1)})$ if the numbers of the sampling points for the $(l+1)$-th and $l$-th layers are both $n$.

By applying Eq. (3) in a multi-layer network, we construct the network structure in a top-down manner: sampling the neighbours of each node in the current layer recursively (see Figure 2 (a)). However, such *node-wise sampling* is still computationally expensive for deep networks, because the number of the nodes to be sampled grows exponentially with the number of layers. Taking a network with depth $d$ for example, the number of sampling nodes in the input layer will increase to $O(n^d)$, leading to significant computational burden for large $d^2$.

**Layer-Wise Sampling.** We equivalently transform Eq. (2) to the following form by applying importance sampling, *i.e.*,

$$h^{(l+1)}(v_i) = \sigma_{W^{(l)}}(N(v_i)\mathbb{E}_{q(u_j|v_1,\cdots,v_n)}[\frac{p(u_j|v_i)}{q(u_j|v_1,\cdots,v_n)}h^{(l)}(u_j)]), \qquad (4)$$

where $q(u_j|v_1,\cdots,v_n)$ is defined as the probability of sampling $u_j$ given all the nodes of the current layer (*i.e.*, $v_1,\cdots,v_n$). Similarly, we can speed up Eq. (4) by approximating the expectation with the Monte-Carlo mean, namely, computing $h^{(l+1)}(v_i) = \sigma_{W^{(l)}}(N(v_i)\hat{\mu}_q(v_i))$ with

$$\hat{\mu}_q(v_i) = \frac{1}{n}\sum_{j=1}^{n} \frac{p(\hat{u}_j|v_i)}{q(\hat{u}_j|v_1,\cdots,v_n)}h^{(l)}(\hat{u}_j), \quad \hat{u}_j \sim q(\hat{u}_j|v_1,\cdots,v_n). \qquad (5)$$

We term the sampling in Eq. (5) as the *layer-wise sampling* strategy. As opposed to the node-wise method in Eq. (3) where the nodes $\{\hat{u}_j\}_{j=1}^{n}$ are generated for each parent $v_i$ independently, the sampling in Eq. (5) is required to be performed only once. Besides, in the node-wise sampling, the neighborhoods of each node are not visible to other parents; while for the layer-wise sampling all sampling nodes $\{\hat{u}_j\}_{j=1}^{n}$ are shared by all nodes of the current layer. This sharing property is able to enhance the message passing at utmost. More importantly, the size of each layer is fixed to $n$, and the total number of sampling nodes only grows linearly with the network depth.

## 4.2 Explicit Variance Reduction

The remaining question for the layer-wise sampling is *how to define the exact form of the sampler* $q(u_j|v_1,\cdots,v_n)$. Indeed, a good estimator should reduce the variance caused by the sampling process, since high variance probably impedes efficient training. For simplicity, we concisely denote the distribution $q(u_j|v_1,\cdots,v_n)$ as $q(u_j)$ below.

According to the derivations of importance sampling in [23], we immediately conclude that

**Proposition 1.** *The variance of the estimator $\hat{\mu}_q(v_i)$ in Eq. (5) is given by*

$$\mathrm{Var}_q(\hat{\mu}_q(v_i)) = \frac{1}{n}\mathbb{E}_{q(u_j)}[\frac{(p(u_j|v_i)|h^{(l)}(u_j)| - \mu_q(v_i)q(u_j))^2}{q^2(u_j)}]. \qquad (6)$$

*The optimal sampler to minimize the variance $\mathrm{Var}_{q(u_j)}(\hat{\mu}_q(v_i))$ in Eq. (6) is given by*

$$q^*(u_j) = \frac{p(u_j|v_i)|h^{(l)}(u_j)|}{\sum_{j=1}^{N} p(u_j|v_i)|h^{(l)}(u_j)|}. \qquad (7)$$

Unfortunately, it is infeasible to compute the optimal sampler in our case. By its definition, the sampler $q^*(u_j)$ is computed based on the hidden feature $h^{(l)}(u_j)$ that is aggregated by its neighborhoods in previous layers. However, under our top-down sampling framework, the neural units of lower layers are unknown unless the network is completely constructed by the sampling.

To alleviate this chicken-and-egg dilemma, we learn a self-dependent function of each node to determine its importance for the sampling. Let $g(x(u_j))$ be the self-dependent function computed based on the node feature $x(u_j)$. Replacing the hidden function in Eq. (7) with $g(x(u_j))$ arrives at

$$q^*(u_j) \quad = \quad \frac{p(u_j|v_i)|g(x(u_j))|}{\sum_{j=1}^{N} p(u_j|v_i)|g(x(u_j))|}, \tag{8}$$

The sampler by Eq. (8) is node-wise and varies for different $v_i$. To make it applicable for the layer-wise sampling, we summarize the computations over all nodes $\{v_i\}_{i=1}^{n}$, thus we attain

$$q^*(u_j) \quad = \quad \frac{\sum_{i=1}^{n} p(u_j|v_i)|g(x(u_j))|}{\sum_{j=1}^{N} \sum_{i=1}^{n} p(u_j|v_i)|g(x(v_j))|}. \tag{9}$$

In this paper, we define $g(x(u_j))$ as a linear function *i.e.* $g(x(u_j)) = W_g x(u_j)$ parameterized by the matrix $W_g \in \mathbb{R}^{1 \times D}$. Computing the sampler in Eq. (9) is efficient, since computing $p(u_j|v_i)$ (*i.e.* the adjacent value) and the self-dependent function $g(x(u_j))$ is fast.

Note that applying the sampler given by Eq. (9) not necessarily results in a minimal variance. To fulfill variance reduction, we add the variance to the loss function and explicitly minimize the variance by model training. Suppose we have a mini-batch of data pairs $\{(v_i, y_i)\}_{i=1}^{n}$, where $v_i$ is the target nodes and $y_i$ is the corresponded ground-true label. By the layer-wise sampling (Eq. (9)), the nodes of previous layer are sampled given $\{v_i\}_{i=1}^{n}$, and this process is recursively called layer by layer until we reaching the input domain. Then we perform a bottom-up propagation to compute the hidden features and obtain the estimated activation for node $v_i$, *i.e.* $\hat{\mu}_q(v_i)$. Certain nonlinear and soft-max functions are further added on $\hat{\mu}_q(v_i)$ to produce the prediction $\bar{y}(\hat{\mu}_q(v_i))$. By taking the classification loss and variance (Eq. (6)) into account, we formulate a hybrid loss as

$$\mathcal{L} = \frac{1}{n} \sum_{i=1}^{n} \mathcal{L}_c(y_i, \bar{y}(\hat{\mu}_q(v_i))) + \lambda \mathrm{Var}_q(\hat{\mu}_q(v_i))), \tag{10}$$

where $\mathcal{L}_c$ is the classification loss (*e.g.*, the crossing entropy); $\lambda$ is the trade-off parameter and fixed as 0.5 in our experiments. Note that the activations for other hidden layers are also stochastic, and the resulting variances should be reduced. In Eq. (10) we only penalize the variance of the top layer for efficient computation and find it sufficient to deliver promising performance in our experiments.

To minimize the hybrid loss in Eq. (10), it requires to perform gradient calculations. For the network parameters, *e.g.* $W^{(l)}$ in Eq. (2), the gradient calculation is straightforward and can be easily derived by the automatically-differential platform, *e.g.*, TensorFlow [24]. For the parameters of the sampler, *e.g.* $W_g$ in Eq. (9), calculating the gradient is nontrivial as the sampling process (Eq. (5)) is non-differential. Fortunately, we prove that the gradient of the classification loss with respect to the sampler is zero. We also derive the gradient of the variance term regarding the sampler, and detail the gradient calculation in the supplementary material

## 5 Preserving Second-Order Proximities by Skip Connections

The GCN update in Eq. (1) only aggregates messages passed from 1-hop neighborhoods. To allow the network to better utilize information across distant nodes, we can sample the multi-hop neighborhoods for the GCN update in a similar way as the random walk [6, 10]. However, the random walk requires extra sampling to obtain distant nodes which is computationally expensive for dense graphs. In this paper, we propose to propagate the information over distant nodes via skip connections.

The key idea of the skip connection is to reuse the nodes of the $(l-1)$-th layer to preserve the second-order proximity (see the definition in [7]). For the $(l+1)$-th layer, the nodes of the $(l-1)$-th layer are actually the 2-hop neighborhoods. If we further add a skip connection from the $(l-1)$-th to the $(l+1)$-th layer, as illustrated in Figure 2 (c), the aggregation will involve both the 1-hop and 2-hop neighborhoods. The calculations along the skip connection are formulated as

$$h_{skip}^{(l+1)}(v_i) \quad = \quad \sum_{j=1}^{n} \hat{a}_{skip}(v_i, s_j) h^{(l-1)}(s_j) W_{skip}^{(l-1)}, \quad i = 1, \cdots, n, \tag{11}$$

where $s = \{s_j\}_{j=1}^n$ denote the nodes in the $(l-1)$-th layer. Due to the 2-hop distance between $v_i$ and $s_j$, the weight $\hat{a}_{skip}(v_i, s_j)$ is supposed to be the element of $\hat{A}^2$. Here, to avoid the full computation of $\hat{A}^2$, we estimate the weight with the sampled nodes of the $l$-th layer, *i.e.*,

$$\hat{a}_{skip}(v_i, s_j) \approx \sum\nolimits_{k=1}^n \hat{a}(v_i, u_k)\hat{a}(u_k, s_j). \tag{12}$$

Instead of learning a free $W_{skip}^{(l-1)}$ in Eq. (11), we decompose it to be

$$W_{skip}^{(l-1)} = W^{(l-1)}W^{(l)}, \tag{13}$$

where $W^{(l)}$ and $W^{(l-1)}$ are the filters of the $l$-th and $(l-1)$-th layers in original network, respectively. The output of skip-connection will be added to the GCN layer (Eq.(1)) before nonlinearity.

By the skip connection, the second-order proximity is maintained without extra 2-hop sampling. Besides, the skip connection allows the information to pass between two distant layers thus enabling more efficient back-propagation and model training.

While the designs are similar, our motivation of applying the skip connection is different to the residual function in ResNets [1]. The purpose of employing the skip connection in [1] is to gain accuracy by increasing the network depth. Here, we apply it to preserve the second-order proximity. In contrast to the identity mappings used in ResNets, the calculation along the skip-connection in our model should be derived specifically (see Eq. (12) and Eq. (13)).

## 6 Discussions and Extensions

**Relation to other sampling methods.** We contrast our approach with GraphSAGE [3] and FastGC-N [21] regarding the following aspects:

1. The proposed layer-wise sampling method is novel. GraphSAGE randomly samples a fixed-size neighborhoods of each node, while FastGCN constructs each layer independently according to an identical distribution. As for our layer-wise approach, the nodes in lower layers are sampled conditioned on the upper ones, which is capable of capturing the between-layer correlations.

2. Our framework is general. Both GraphSAGE and FastGCN can be categorized as the specific variants of our framework. Specifically, the GraphSAGE model is regarded as a node-wise sampler in Eq (3) if $p(u_j|v_i)$ is defined as the uniform distribution; FastGCN can be considered as a special layer-wise method by applying the sampler $q(u_j)$ that is independent to the nodes $\{v_i\}_{i=1}^n$ in Eq. (5).

3. Our sampler is parameterized and trainable for explicit variance reduction. The sampler of GraphSAGE or FastGCN involves no parameter and is not adaptive for minimizing variance. In contrast, our sampler modifies the optimal importance sampling distribution with a self-dependent function. The resulting variance is explicitly reduced by fine-tuning the network and sampler.

**Taking the attention into account.** The GAT model [13] applies the idea of self-attention to graph representation learning. Concisely, it replaces the re-normalization of the adjacency matrix in Eq. (1) with specific attention values, *i.e.*, $h^{(l+1)}(v_i) = \sigma(\sum_{j=1}^N a((h^{(l)}(v_i), (h^{(l)}(u_j))h^{(l)}(v_j)W^{(l)})$, where $a(h^{(l)}(v_i), h^{(l)}(u_j))$ measures the attention value between the hidden features $v_i$ and $u_j$, which is derived as $a(h^{(l)}(v_i), h^{(l)}(u_j)) = \text{SoftMax}(\text{LeakyReLU}(W_1 h^{(l)}(v_i), W_2 h^{(l)}(u_j)))$ by using the LeakyReLU nonlinearity and SoftMax normalization with parameters $W_1$ and $W_2$.

It is impracticable to apply the GAT-like attention mechanism directly in our framework, as the probability $p(u_j|v_i)$ in Eq. (9) will become related to the attention value $a(h^{(l)}(v_i), h^{(l)}(u_j))$ that is determined by the hidden features of the $l$-th layer. As discussed in § 4.2, computing the hidden features of lower layers is impossible unless the network is already built after sampling. To solve this issue, we develop a novel attention mechanism by applying the self-dependent function similar to Eq. (9). The attention is computed as

$$a(x(v_i), x(u_j)) = \frac{1}{n}\text{ReLu}(W_1 g(x(v_i)) + W_2 g(x(u_j))), \tag{14}$$

where $W_1$ and $W_2$ are the learnable parameters.

## 7 Experiments

We evaluate the performance of our methods on the following benchmarks: (1) categorizing academic papers in the citation network datasets–Cora, Citeseer and Pubmed [11]; (2) predicting which

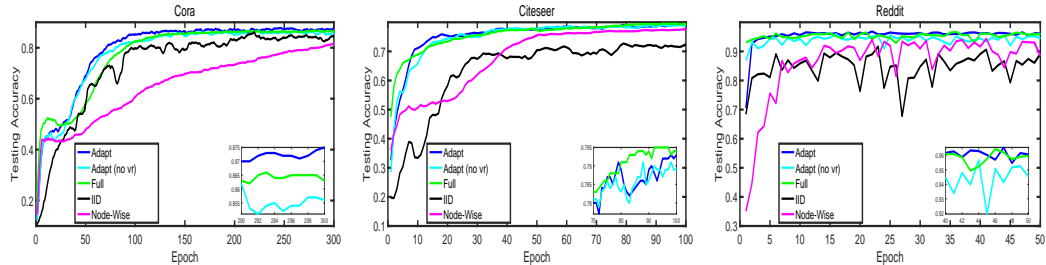

Figure 2: The accuracy curves of test data on Cora, Citeseer and Reddit. Here, one training epoch means a complete pass of all training samples.

community different posts belong to in Reddit [3]. These graphs are varying in sizes from small to large. Particularly, the number of nodes in Cora and Citeseer are of scale $O(10^3)$, while Pubmed and Reddit contain more than $10^4$ and $10^5$ vertices, respectively. Following the supervised learning scenario in FastGCN [21], we use all labels of the training examples for training. More details of the benchmark datasets and more experimental evaluations are presented in the supplementary material.

Our sampling framework is *inductive* in the sense that it clearly separates out test data from training. In contrast to the *transductive* learning where all vertices should be provided, our approach aggregates the information from each node's neighborhoods to learn structural properties that can be generalized to unseen nodes. For testing, the embedding of a new node may be either computed by using the full GCN architecture or approximated through sampling as is done in model training. Here we use the full architecture as it is more straightforward and easier to implement. For all datasets, we employ the network with two hidden layers as usual. The hidden dimensions for the citation network datasets (*i.e.*, Cora, Citeseer and Pubmed) are set to be 16. For the Reddit dataset, the hidden dimensions are selected to be 256 as suggested by [3]. The numbers of the sampling nodes for all layers excluding the top one are set to 128 for Cora and Citeseer, 256 for Pubmed and 512 for Reddit. The sizes of the top layer (*i.e.* the stochastic mini-batch size) are chosen to be 256 for all datasets. We train all models using early stopping with a window size of 30, as suggested by [9], and report the results corresponding to the best validation accuracies. Further details on the network architectures and training settings are contained in the supplementary material.

### 7.1 Alation Studies on the Adaptive Sampling

**Baselines.** The codes of GraphSAGE [3] and FastGCNN [21] provided by the authors are implemented inconsistently; here we re-implement them based on our framework to make the comparisons more fair[3]. In details, we implement the GraphSAGE method by applying the node-wise strategy with a uniform sampler in Eq. (3), where the number of the sampling neighborhoods for each node are set to 5. For FastGCN, we adopt the Independent-Identical-Distribution (IID) sampler proposed by [21] in Eq. (5), where the number of the sampling nodes for each layer is the same as our method. For consistence, the re-implementations of GraphSAGE and FastGCN are named as *Node-Wise* and *IID* in our experiments. We also implement the *Full* GCN architecture as a strong baseline. All compared methods shared the same network structure and training settings for fair comparison. We have also conducted the attention mechanism introduced in § 6 for all methods.

**Comparisons with other sampling methods.** The random seeds are fixed and no early stopping is used for the experiments here. Figure 2 reports the converging behaviors of all compared methods during training on Cora, Citeseer and Reddit[4]. It demonstrates that our method, denoted as *Adapt*, converges faster than other sampling counterparts on all three datasets. Interestingly, our method even outperforms the *Full* model on Cora and Reddit. Similar to our method, the *IID* sampling is also layer-wise, but it constructs each layer independently. Thanks to the conditional sampling, our method achieves more stable convergent curve than the *IID* method as Figure 2 shown. It turns out that considering the between-layer information helps in stability and accuracy.

Moreover, we draw the training time in Figure 3 (a). Clearly, all sampling methods run faster than the *Full* model. Compared to the *Node-Wise* method, our approach exhibits a higher training speed due to

Table 1: Accuracy Comparisons with state-of-the-art methods.

| Methods | Cora | Citeseer | Pubmed | Reddit |
|---|---|---|---|---|
| KLED [25] | 0.8229 | - | 0.8228 | - |
| 2-hop DCNN [18] | 0.8677 | - | 0.8976 | - |
| FastGCN [21] | 0.8500 | 0.7760 | 0.8800 | 0.9370 |
| GraphSAGE[3] | 0.8220 | 0.7140 | 0.8710 | 0.9432 |
| Full | $0.8664 \pm 0.0011$ | $0.7934 \pm 0.0026$ | $0.9022 \pm 0.0008$ | $0.9568 \pm 0.0069$ |
| IID | $0.8506 \pm 0.0048$ | $0.7387 \pm 0.0078$ | $0.8200 \pm 0.0114$ | $0.8611 \pm 0.0437$ |
| Node-Wise | $0.8202 \pm 0.0133$ | $0.7734 \pm 0.0081$ | $0.9002 \pm 0.0017$ | $0.9449 \pm 0.0026$ |
| Adapt (no vr) | $0.8588 \pm 0.0062$ | $0.7942 \pm 0.0022$ | $0.9060 \pm 0.0024$ | $0.9501 \pm 0.0047$ |
| **Adapt** | $\mathbf{0.8744 \pm 0.0034}$ | $\mathbf{0.7966 \pm 0.0018}$ | $\mathbf{0.9060 \pm 0.0016}$ | $\mathbf{0.9627 \pm 0.0032}$ |

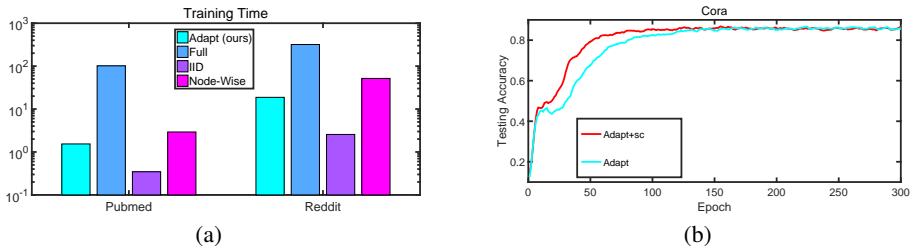

(a)  (b)

Figure 3: (a) Training time per epoch on Pubmed and Reddit. (b) Accuracy curves of testing data on Cora for our Adapt method and its variant by adding skip connections.

the more compact architecture. To show this, suppose the number of nodes in the top layer is $n$, then for the *Node-Wise* method the input, hidden and top layers are of sizes $25n$, $5n$ and $n$, respectively, while the numbers of the nodes in all layers are $n$ for our model. Even with less sampling nodes, our model still surpasses the *Node-Wise* method by the results in Figure 2.

**How important is the variance reduction?** To justify the importance of the variance reduction, we implement a variant of our model by setting the trade-off parameter as $\lambda = 0$ in Eq. (10). By this, the parameters of the self-dependent function are randomly initialized and no training is performed. Figure 2 shows that, removing the variance loss does decrease the accuracies of our method on Cora and Reddit. For Citeseer, the effect of removing the variance reduction is not so significant. We conjecture that the average degree of Citeseer (*i.e.* 1.4) is smaller than Cora (*i.e.* 2.0) and Reddit (*i.e.* 492), and penalizing the variance is not so impeding due to the limited diversity of neighborhoods.

**Comparisons with other state-of-the-art methods.** We contrast the performance of our methods with the graph kernel method KLED [25] and Diffusion Convolutional Network (DCN) [18]. We use the reported results of KLED and DCN on Cora and Pubmed in [18]. We also summarize the results of GraphSAGE and FastGCN by their original implementations. For GraphSAGE, we report the results by the mean aggregator with the default parameters. For FastGCN, we directly make use of the provided results by [21]. For the baselines and our approach, we run the experiments with random seeds over 20 trials and record the mean accuracies and the standard variances. All results are organized in Table 1. As expected, our method achieves the best performance among all datasets, which are consistent with the results in Figure 2. It is also observed that removing the variance reduction will decrease the performance of our method especially on Cora and Reddit.

## 7.2 Evaluations of the Skip Connection

We evaluate the effectiveness of the skip connection on Cora. For the experiments on other datasets, we present the details in the supplementary material. The original network has two hidden layers. We further add a skip connection between the input and top layers, by using the computations in Eq. (12) and Eq. (13). Figure 2 displays the convergent curves of the original *Adapt* method and its variant with the skip connection, where the random seeds are shared and no early stopping is adapted. Although the improvement by our skip connection is not big regarding the final accuracy, it indeed speeds up the convergence significantly. This can be observed from Figure 3 (b) where adding the skip connection reduces the required epochs to converge from around 150 to 100.

Table 2: Testing Accuracies on Cora.

| Adapt | Adapt+sc | Adapt+2-hop |
|---|---|---|
| $0.8744 \pm 0.0034$ | $0.8774 \pm 0.0032$ | $0.8814 \pm 0.0017$ |

We run experiments with different random seeds over 20 trials and report the mean results obtained by early stopping in Table 2. It is observed that the skip connection slightly improves the performance. Besides, we explicitly involve the 2-hop neighborhood sampling in our method by replacing the re-normalization matrix $\hat{A}$ with its 2-order power expansion, *i.e.* $\hat{A} + \hat{A}^2$. As displayed in Table 2, the explicit 2-hop sampling further boosts the classification accuracy. Although the skip-connection method is slightly inferior to the explicit 2-hop sampling, it avoids the computation of (*i.e.* $\hat{A}^2$) and yields more computationally beneficial for large and dense graphs.

## 8   Conclusion

We present a framework to accelerate the training of GCNs through developing a sampling method by constructing the network layer by layer. The developed layer-wise sampler is adaptive for variance reduction. Our method outperforms the other sampling-based counterparts: GraphSAGE and FastGCN in effectiveness and accuracy on extensive experiments. We also explore how to preserve the second-order proximity by using the skip connection. The experimental evaluations demonstrate that the skip connection further enhances our method in terms of the convergence speed and eventual classification accuracy.

## Footnotes

[1]Here, lower layers denote the ones closer to the input.

[2]One can reduce the complexity of the node-wise sampling by removing the repeated nodes. Even so, for dense graphs, the sampling nodes will still quickly fills up the whole graph as the depth grows.

[3]We also perform experimental comparisons by using the public codes of FastGCN in the supplementary material.

[4]The results on Pubmed are provided in the supplementary material.

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
