[Supplementary Material · supplementary_material.pdf]

# Supplementary Material to Adaptive Sampling Towards Fast Graph Representation Learning

**Wenbing Huang**[1]**, Tong Zhang**[2]**, Yu Rong**[1]**, Junzhou Huang**[1]
[1] Tencent AI Lab. ;
[2] Australian National University;
hwenbing@126.com, tong.zhang@anu.edu.au
yu.rong@hotmail.com, joehhuang@tencent.com

This supplementary material provides the gradient calculation of the loss function ( Eq. (10)) with respect to the sampler. It also contains more setting details and more results for the experiments.

## 1 Gradient Calculation

We prove that the gradient of the expectation $\hat{\mu}_q(v_i)$ in Eq. (5) with respect to the sampler $q(u_j)$ is equal to zero. To demonstrate this, we decompose the gradient as

$$
\begin{aligned}
\frac{\partial}{\partial q(u_j)}\hat{\mu}_q(v_i) &= \frac{\partial}{\partial q(u_j)}\mathbb{E}_{q(u_j)}\big[\frac{p(u_j|v_i)}{q(u_j)}h^{(l)}(u_j)\big], \\
&= \frac{\partial}{\partial q(u_j)}\mathbb{E}_{p(u_j|v_i))}[h^{(l)}(u_j)], \\
&= 0.
\end{aligned}
\tag{15}
$$

Hence, the gradient of the classification loss in Eq. (10) regarding the sampler is equal to zero. To perform the gradient calculation for the variance term, we first estimate it with the sampled instances by

$$
\widehat{\mathrm{Var}}_q(\hat{\mu}_q(v_i)) = \frac{1}{n^2}\sum\nolimits_{j=1}^{n}\frac{\big(p_\theta(\hat{u}_j|v_i)|h^{(l)}(\hat{u}_j)| - \hat{\mu}_q(v_i)q(\hat{u}_j)\big)^2}{q^2(\hat{u}_j)},
\tag{16}
$$

whose gradient is given by

$$
\frac{\partial}{\partial q(\hat{u}_j)}\widehat{\mathrm{Var}}_q(\hat{\mu}_q(v_i))) = -\frac{1}{n^2}\frac{p_\theta(\hat{u}_j|v_i)|h^{(l)}(\hat{u}_j)|\big(p_\theta(\hat{u}_j|v_i)|h^{(l)}(\hat{u}_j)| - \hat{\mu}_q(v_i)q(\hat{u}_j)\big)}{q^3(\hat{u}_j)},
\tag{17}
$$

where the samples $\{\hat{u}_j\}_{j=1}^{n}$ generated from $q(u_j)$ independently.

## 2 More Experimental Evaluations

**Datasets.** The Cora, Citeseer and Pubmed datasets are downloaded from `https://github.com/tkipf/gcn`. We follow the setting as [1] by keeping the validation and test indexes unchanged but using all remaining samples for training. The Reddit dataset is from `http://snap.stanford.edu/graphsage/`. The statistics of four datasets are summarized in Table 3.

**Further implementation details.** The initial learning rates for the Adam optimizer are set to be 0.001 for Cora, Citeseer and Pubmed, and 0.01 for Reddit. The weight decays for all datasets are selected to be 0.0004. We apply ReLu function as the activation function and no dropout in our

Table 3: Dataset Statistics.

| Datasets | Nodes | Edges | Classes | Features | Training/Validation/Testing |
|----------|-------|-------|---------|----------|-----------------------------|
| Cora | 2,708 | 5,429 | 7 | 1,433 | 1, 208/500/1,000 |
| Citeseer | 3,327 | 4,732 | 6 | 3,703 | 1, 812/500/1,000 |
| Pubmed | 19,717 | 44,338 | 3 | 500 | 18, 217/500/1,000 |
| Reddit | 232,965 | 11,606,919 | 41 | 602 | 152,410/23,699/55,334 |

experiments. As presented in the paper, all models are implemented with 2-hidden-layer networks. For the Reddit dataset, we follow the suggestion by [1] to fix the weight of the bottom layer and pre-compute the product $\hat{A}H^{(0)}$ given the input features for efficiency. All experiments are conducted on a single Tesla P40 GPU. We apply the early-stopping for the training with a window size of 30 and apply the model that achieves the best validation accuracy for testing.

**More results on the variance reduction.** As shown in Table 1, it is sufficient to boost the performance by only reducing the variance of the top layer. Indeed, it is convenient to reduce the variances of all layers in our method, e.g., adding them all to the loss. To show this, we conduct an experiment on Cora by minimizing the variances of both the first and top hidden layers, where the experimental settings are the same as Table 1. The result is $0.8780 \pm 0.0014$, which slightly outperforms the original accuracy in Table 1 (i.e. $0.8744 \pm 0.0034$).

**Comparisons with FastGCN by using the official codes.** We use the public code to re-run the experiments of FastGCN in Figure 2 and Table 1. The average accuracies of FastGCN for four datasets are $0.840 \pm 0.005$, $0.774 \pm 0.004$, $0.881 \pm 0.002$ and $0.920 \pm 0.005$. The running curves of Figure 2 in the paper are updated by Figure 5 here. Clearly, our method still outperforms FastGCN remarkably. We have observed the inconsistences between the official implementations of GraphSAGE and FastGraph including the adjacent matrix construction, hidden dimensions, mini-batch sizes, maximal training epoches and other engineering tricks not mentioned in their papers. For fair comparisons, we re-implements them and uses the same experimental settings as our method in the paper.

Figure 4: The accuracy curves of test data on Cora, Citeseer and Reddit. Here, one training epoch means a complete pass of all training samples. The sampling nodes of each hidden layer for FastGCN and our method on Cora, Citeseer, Pubmed and Reddit are selected as 128, 128, 256, and 512, respectively.

**More results on Pubmed.** In the paper, Figure 2 displays the accuracy curves of test data on Cora, Citeseer and Reddit, where the random seeds are fixed. For those on Pubmed, we provide results in Figure 5. Obviously, our method outperforms the IID and Node-Wise counterparts consistently. The Full model achieves the best accuracy around the 30-th epoch, but drops down after the 60-th epoch properly due to the overfitting. In contrast, our performance is more stable and it gives even better results in the end. Performing the variance reduction on this dataset is only helpful during the early stage, but contributes little when the model converges.

Table 3 (b) reports the accuracy curve of the model with the skip connection on Cora. Here, we evaluate the effectiveness of the skip connection on Citeseer and Pubmed in Figure 6. It demonstrates that the skip connection is helpful to speed up the convergence on Citeseer. While on the Pubmed dataset, adding the skip connection boosts the performance only during early training epochs. For the Reddit dataset, we can not apply the skip connection in the network since the bottom layer is fixed and the output features are pre-computed.

Figure 5: The accuracy curves of test data on Pubmed. Here, a training epoch means a complete pass of all training samples.

Figure 6: Accuracy curves of testing data on Citeseer and Pubmed for our Adapt method and its variant by adding skip connections.

## References

[1] Jie Chen, Tengfei Ma, and Cao Xiao. Fastgcn: Fast learning with graph convolutional networks via importance sampling. *arXiv preprint arXiv:1801.10247*, 2018.