[Reviews · NeurIPS 2018]

Reviewer 1



Post-rebuttal: I have adjusted the score by raising one point, if the authors incorporate all the changes promised, including the clarification on skip connection, the experiments on attention, and the correction of inconsistent results. My evaluation of the overall paper remains that the experimental results are convincing but the contribution seems incremental. ================================================= This paper proposes two ingredients to improve the performance of GCN: (1) an adaptive sampling approach to addressing scalability and (2) a skip connection approach to improving classification accuracy. The adaptive sampling is an extension of FastGCN. The innovative point is that the proposal distribution is learned through variance reduction. A slight drawback is that the loss function includes the variance part for only the top layer. The skip connection is reasonable, but judged from the experimental results, this approach does not seem to offer noticeable advantage. The authors draw a connection with GAT, which is interesting. However, it is unclear how this part is connected to the rest of the paper. For example, the authors seem to propose an alternative formula for computing the attention (Equation (14)), but no experiments are presented. The self-implementation results in Table 1 are not quite consistent with the published ones from GraphSAGE and FastGCN. Could the authors explain the difference? Would the inconsistency be caused by hyperparameter tuning? The writing and language could be improved. Word usage, such as "visitable" and "impracticable", should be proofread. Overall, while the experiments confirm that the proposed ideas work, the reader feels that the contribution is incremental.

Reviewer 2



The paper proposed a new adaptive sampling scheme for graph convolutional networks and gained extremely good results on several benchmarks. The work is an extension of GraphSAGE and FastGCN. Different from GraphSAGE which samples fixed-size neighborhoods for each node, the proposed method sample nodes layer-wise. FastGCN is the most related work, which has the same ideas of layer-wise sampling and variance induction. The improvement of the proposed method on FastGCN mainly lies on the following aspects: 1. The proposed method sample nodes conditioned on the samples of previous layers instead of using layer-independent sampling scheme in FastGCN. 2. For computational efficiency, FastGCN removed the hidden feature part in the optimal sampling distribution in variance reduction, the proposed method used a linear function of node features instead and add the variance in the final objective function. 3. Adding skip connection to preserve second-order proximities. The paper is well written and technically correct. The ideas of layer-wise sampling and variance reduction are not new, and the improvement on FastGCN is not very significant. But the layer-dependent sampling is interesting, and it does solve the potential drawbacks of FastGCN and leads to a significant performance gain. So the technical contribution is acceptable. The experimental results are promising. However, the running curve of FastGCN is quite different from the original implementation. I directly run the public FastGCN codes on one dataset (pubmed), the running curve is consistent with the original paper. As there are public codes for FastGCN and GraphSAGE, I think it is better to use their original codes. Especially for FastGCN, as there is a performance gap, I suggest the authors re-check their own implementation or use original FastGCN codes instead. If the paper is accepted, I hope the authors could modify the running curves in Figure2. It is better to add another recent sampling method in the related work: Chen, Jianfei, Jun Zhu, and Le Song. "Stochastic Training of Graph Convolutional Networks with Variance Reduction." International Conference on Machine Learning. 2018.

Reviewer 3



In this paper, the authors solve the scalability issue of GCNs by developing a layer-by-layer sampling method for network construction. There are two benefits for this: first, it enables the neighborhood sharing in upper layers; second, the parameters of the sampler and the network can be finetuned to reduce the variance explicitly. Moreover, to enable message passing across distant nodes, this paper proposes to apply skip connections to reuse to nodes in the (l-1)-th layer as the 2-hop neighborhoods of the (l+1)-th layer. Finally, remarkable improvements in accuracy and speedups are demonstrated for the proposed methods on four benchmarks. 1) Clarity: This is a Well-written paper. It is well organized and easy to understand. 2) Novelty: Compared to GraphSage and FastGCN that have been developed previously, the layer-wise sampling scheme by this paper is novel and efficient. Also, the idea of applying skip-connections to preserve second-order proximity is interesting. 3) Significance: Experimental evaluations verify the fast convergence speed of the proposed method compared to other sampling counter-parts. The importance of the variance reduction is also justified by Table 1. 4) Minor things: - Line 252 in the paper says the window-size of early stopping is 10, while in the supplementary the number becomes 30 in line 22. Which is correct? - To justify the importance of the variance reduction, the authors set \lambda=0. However, without the variance term, how can we train the parameters of the self-dependent function, since the gradient of the classification loss is always zero? - In Line 120, the computational complexity for Eq.(1) should be O(|E|D^lD^{l-1}), not O(N^2 D^lD^{l-1}). In all, a rather thorough paper that derives an efficient way to perform convolutions on graphs using layer-wise sampling and skip-connections. At one hand, this leads to improvements over other competing methods in terms of accuracy and convergence speed. Hence, I recommend accepting this paper. ###################After Rebuttal########################### I think the authors have addressed my previous concerns, which make me not to change my mind and recommend the acceptance.